# Extraction of Interconnect Parasitic Capacitance Matrix Based on Deep Neural Network

Yaoyao Ma [1,2,3], Xiaoyu Xu [4], Shuai Yan [4], Yaxing Zhou [4], Tianyu Zheng [4], Zhuoxiang Ren [4,5,*] and Lan Chen [1,3]

1   Institute of Microelectronics of the Chinese Academy of Sciences, Beijing 100029, China
2   University of Chinese Academy of Sciences, Beijing 100049, China
3   Beijing Key Laboratory of Three-Dimensional and Nanometer Integrated Circuit Design Automation Technology, Beijing 100029, China
4   Institute of Electrical Engineering, Chinese Academy of Sciences, Beijing 100190, China
5   Group of Electrical and Electronic Engineering of Paris, Sorbonne Université, Université Paris-Saclay, CentraleSupélec, CNRS, 75005 Paris, France
*   Correspondence: zhuoxiang.ren@upmc.fr

**Abstract:** Interconnect parasitic capacitance extraction is crucial in analyzing VLSI circuits' delay and crosstalk. This paper uses the deep neural network (DNN) to predict the parasitic capacitance matrix of a two-dimensional pattern. To save the DNN training time, the neural network's output includes only coupling capacitances in the matrix, and total capacitances are obtained by summing corresponding predicted coupling capacitances. In this way, we can obtain coupling and total capacitances simultaneously using a single neural network. Moreover, we introduce a mirror flip method to augment the datasets computed by the finite element method (FEM), which doubles the dataset size and reduces data preparation efforts. Then, we compare the prediction accuracy of DNN with another neural network ResNet. The result shows that DNN performs better in this case. Moreover, to verify our method's efficiency, the total capacitances calculated from the trained DNN are compared with the network (named DNN-2) that takes the total capacitance as an extra output. The results show that the prediction accuracy of the two methods is very close, indicating that our method is reliable and can save the training workload for the total capacitance. Finally, a solving efficiency comparison shows that the average computation time of the trained DNN for one case is not more than 2% of that of FEM.

**Keywords:** interconnect wire; parasitic capacitance matrix; data augmentation; DNN; ResNet

## 1. Introduction

With the shrinking of the feature size, the interconnect wires' delay caused by the parasitic parameters has exceeded the gate delay and becomes the main part of the chip's total delay [1]. Additionally, parasitic parameters of interconnect wires also cause crosstalk between wires, which may lead to confusion in the transmitted signal logic. Parasitic parameters consist of resistance, capacitance, and inductance; among them, parasitic capacitance plays a significant role owing to its extraction complexity and critical influence on signal integrity analysis [2,3].

In advanced process technology nodes, the accuracy requirement of the parasitic capacitance extraction is that the relative error should be less than 5% [4]. The parasitic capacitance extraction methods can be divided into two categories: the field solver method and the pattern-matching method [5]. The field solver method solves maxwell equations through traditional numerical methods, such as floating random walk (FRW) [6], finite element method (FEM) [7], and boundary element method (BEM) [8,9], etc. It has high accuracy but suffers high calculation costs for full chip extraction. The pattern-matching method first segments the layout and obtains the interconnect wires' patterns; then matches

the corresponding pattern in the pattern library; in the end, calculates each pattern's capacitance through pre-built capacitance models that are stored in the pattern library. It is fast and is often used for parasitic extraction at the full chip level. Figure 1 shows the flow of the pattern-matching method. The capacitance extraction dimension changes from 1D, 2D, and 2.5D to 3D and the extraction accuracy is improved accordingly [10]. Directly using the 3D model will cause huge memory and time cost for the full-chip extraction [11]. The 2.5D method is an alleviation of 3D method, which considers the 3D effect by a combination of two orthogonal 2D patterns [12]. It achieves a good balance between accuracy and efficiency and is used in several commercial electronic design automation (EDA) tools, such as StarRC [13] and Calibre xRC [14].

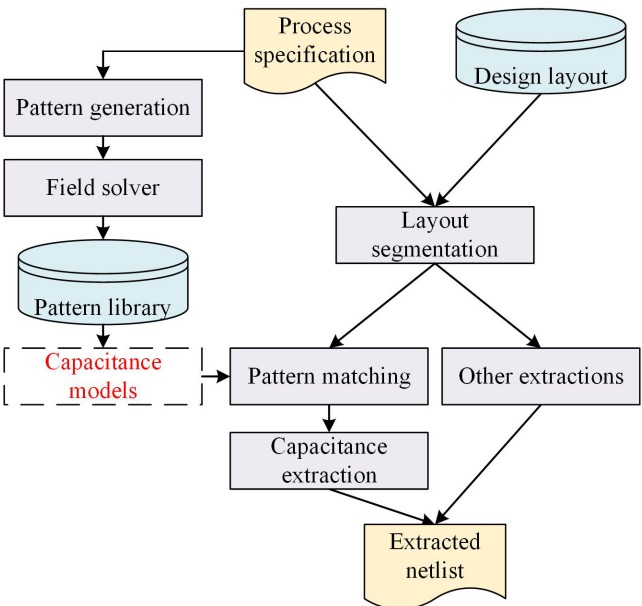

**Figure 1.** The flow of parasitic capacitance extraction using the pattern-matching method [3,15].

With the continuous development of big data technology and high-performance computing, machine learning and deep learning have been applied to many fields [16]. In addition to common fields such as image analysis and natural language processing [17], they have also been implemented in the field of electronic and electrical engineering [18–20]. Some researchers use machine learning and deep learning to solve the problem of parasitic parameter extraction. Kasai et al. [21] employed multilayer perception (MLP) to extract the total capacitance and partial coupling capacitances of 3D interconnect wires. The dimension of the MLP output layer is only one. Hence, it needs to train multiple neural networks to predict the total capacitance and coupling capacitances of the same pattern. Besides, its prediction error for total capacitance was not very good since some errors exceeded 10%. Li et al. [22] combined the adjacent capacitance formulas through the Levenberg–Marquardt least square method, reducing the number of capacitance formulas in the pattern library. It also constructed a classifier for automatic pattern matching through a deep neural network (DNN), which improved pattern-matching efficiency. Yang et al. [11] leveraged a convolutional neural network (CNN), ResNet-34, to extract parasitic capacitances of 2D interconnect wires. For the total capacitance and coupling capacitances, this work trains two separate ResNet networks for prediction. Furthermore, it also compared the prediction performance of the CNN with the MLP, showing that the accuracy is higher than the MLP. Abouelyazid et al. [4] focused on the interconnect pattern containing metal connectivity and trained a DNN model to predict one specific coupling capacitance. Abouelyazid et al. [5] took into account the trapezoidal, wire thickness, dielectric layer thickness, et al. into the model variation and employed DNN and support vector regression (SVN) to predict one specific coupling capacitance of 2D patterns, respectively. Abouelyazid et al. [23]

employed DNN to predict the parasitic capacitance matrix. For total capacitances and coupling capacitances in the matrix, this work utilized two separate DNN networks for prediction. Moreover, a hybrid extraction flow was proposed. It can identify the accuracy of three extraction methods, field-solver, rule-based, and DNN-based, and choose the fastest extraction method to meet the user's required accuracy.

In the parasitic capacitance matrix, the total capacitance on the matrix diagonal equals the sum of the remaining coupling capacitances in the same row. Using this constraint, we train only one DNN model to predict the capacitance matrix in this study. The neural network's output includes only coupling capacitances in the capacitance matrix, and the total capacitances are obtained by summing corresponding predicted coupling capacitances. The trained DNN model can be used as a capacitance model in the pattern-matching method shown in Figure 1. The main contributions of this work are as follows. First, we propose a mirror flip method to realize data augmentation. Second, we use the DNN to predict the parasitic capacitance matrix. The neural network output includes merely coupling capacitances in the matrix. The total capacitance is obtained by calculating the sum of the coupling capacitances. Table 1 shows the comparison of our work with other state-of-the-art works.

**Table 1.** The comparison among different capacitance extraction methods, including our works.

| Property | Abouelyazid et al. [4] | Abouelyazid et al. [5] | Kasai et al. [21] | Yang et al. [11] | Abouelyazid et al. [23] | This Work |
|---|---|---|---|---|---|---|
| Prediction content | One coupling capacitance | One coupling capacitance | One capacitance | One total capacitance and its corresponding coupling capacitances | Capacitance matrix | Capacitance matrix |
| Pattern dimension | 2D | 2D | 3D | 2D | 3D | 2D |
| Data augmentation | No | No | No | No | No | Yes |
| Type of the method | DNN | DNN, SVN | MLP | ResNet | DNN | DNN |
| Number of neural networks | 1 | 1 | 1 | 2 | 2 | 1 |

The rest of the paper is organized as follows. Section 2 first recalls the definition of the parasitic capacitance matrix. Then it introduces the data acquisition flow, data augmentation method, and the DNN structure used in the work. Finally, the density-based data representation for the ResNet input and the ResNet structure used in this work are described in detail. In Section 3, we first utilize DNN to predict the parasitic capacitance matrix of a 2D pattern. Then, ResNet is employed to predict the parasitic capacitance matrix for comparison. Then, to further verify our method, the calculated total capacitances from the trained DNN are compared with another network (named DNN-2) that takes the total capacitance as the output. Then, the prediction performance of DNN under different training set sizes is tested. Then, the solving efficiency of the trained DNN and the FEM is compared. In the end, we apply our method to a new pattern to verify its feasibility. Section 4 discusses the experimental results and future works. Section 5 concludes this work.

## 2. Materials and Methods

### 2.1. Parasitic Capacitance Matrix

For multi-conductor interconnect systems, capacitance extraction means calculating the capacitance matrix between those conductors. The coupling capacitance $C_{ij}$ between conductor $i$ and $j$ satisfies Equation (1). $V_{ij}$ represents the potential difference between $i$ and $j$, and $Q_j$ denotes the total charges on the surface of conductor $j$.

$$C_{ij} = \frac{Q_j}{V_{ij}} \quad (j \neq i) \tag{1}$$

For an electrostatic system containing $n$ conductors, the capacitance matrix is defined as Equation (2), where $q_i$ $(i = 1, 2, \ldots, n)$ denotes the total charges on the conductor $i$. $u_i$ is the potential of conductor $i$. $C_{ii}$ on the matrix diagonal represents the total capacitance or self-capacitance of conductor $i$, whose value is equal to the sum of the remaining coupling capacitances $C_{ij}$ $(j = 1, 2, \ldots, n, \ j \neq i)$ in the same row.

The method to calculate the capacitance matrix is roughly as follows. The potential of conductor $i$ is set as $u_i = 1$, and the potential of other conductors is $u_j = 0$ $(j = 1, 2, \ldots, n, j \neq i)$, then $V_{ij} = 1$. The $C_{ij}$ can be obtained by computing the total charges on the surface of conductor $j$. In addition, since the total capacitance of conductor $i$ is the capacitance of this conductor to the ground, and the potential difference between them is also 1. Hence, the total capacitance $C_{ii}$ equals the charges on the surface of conductor $i$. Repeating this process, the entire capacitance matrix can be determined. Moreover, since the capacitance between two conductors is a single value, the capacitance matrix is symmetric.

$$\begin{bmatrix} q_1 \\ q_2 \\ \vdots \\ q_n \end{bmatrix} = \begin{bmatrix} C_{11} & C_{12} & \ldots & C_{1n} \\ C_{21} & C_{22} & \ldots & C_{2n} \\ \vdots & \vdots & \vdots & \vdots \\ C_{n1} & C_{n2} & \ldots & C_{nn} \end{bmatrix} \begin{bmatrix} u_1 \\ u_2 \\ \vdots \\ u_n \end{bmatrix} \tag{2}$$

*2.2. Dataset Acquisition*

In the pattern-matching method, 2.5D extraction is a way that considers the 3D effect with a combination of two orthogonal 2D patterns. Moreover, each 2D pattern is defined by the combination of different metal layers and the number of conductors on each layer [11]. Figure 2 shows a 2D pattern studied in this work. It contains three metal layers, layer $k$, layer m, layer $n$, and a large substrate ground. Each layer contains one or more interconnect conductors. The substrate ground and conductors are embedded in the dielectric, whose relative permittivity is 3.9. The studied pattern contains five conductors and one substrate ground. Conductor 1 is always in the center, and its central position $x_1$ is a constant, which equals 0. Hence, there are nine geometric variables in total, including the center positions and widths of conductors, namely $x_2$, $x_3$, $x_4$, $x_5$, and $w_1$, $w_2$, $w_3$, $w_4$, $w_5$.

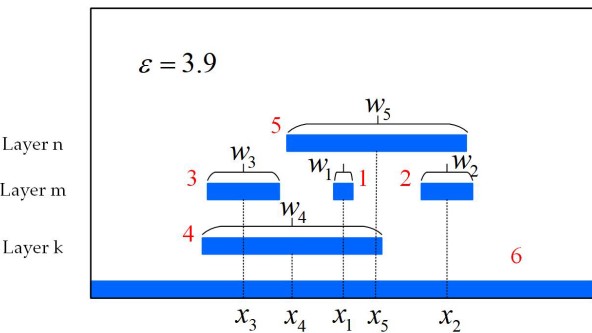

**Figure 2.** The illustration of the studied pattern and its geometry variables.

The capacitance matrix of this pattern is defined in Equation (3). It ignores the substrate ground's total capacitance and only considers the coupling capacitances between the substrate ground and other conductors.

$$C = \begin{bmatrix} C_{11} & C_{12} & C_{13} & C_{14} & C_{15} & C_{16} \\ C_{21} & C_{22} & C_{23} & C_{24} & C_{25} & C_{26} \\ C_{31} & C_{32} & C_{33} & C_{34} & C_{35} & C_{36} \\ C_{41} & C_{42} & C_{43} & C_{44} & C_{45} & C_{46} \\ C_{51} & C_{52} & C_{53} & C_{54} & C_{55} & C_{56} \end{bmatrix} \tag{3}$$

The acquisition of datasets is a crucial step in deep learning. In order to achieve broad coverage of the sampling range, we employ the Latin hypercube sampling (LHS) [24]

method. The studied pattern contains nine geometric variables. Suppose sampling $N$ sets of data for each parameter, the flow of LHS is roughly as follows. First, randomly sample $N$ values within the range of each geometric variable. Then randomly match the sampling results, then a dataset with $N \times 9$ size is obtained. After obtaining the parameter sampling results, we then construct simulation models and utilize the FEM program to compute their capacitance matrix. To sum up, the flow of the dataset acquisition is shown in Figure 3, and the above process is conducted through the FEM software EMPbridge developed by the authors' team [25], whose modeling and solving result interfaces are shown in Figure 4.

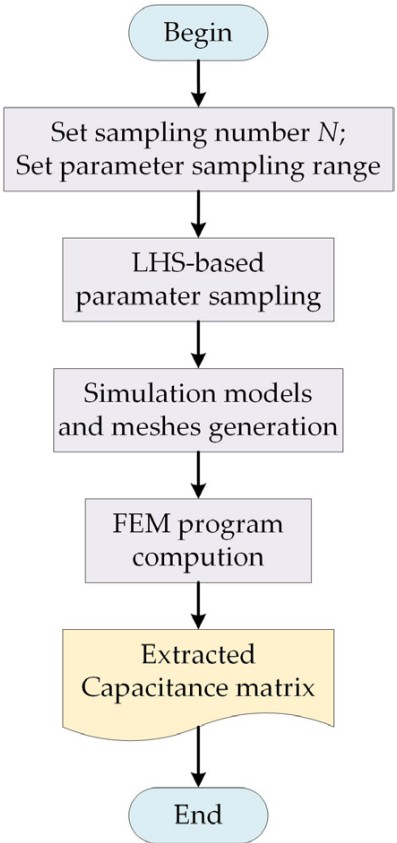

**Figure 3.** The flow of datasets acquisition.

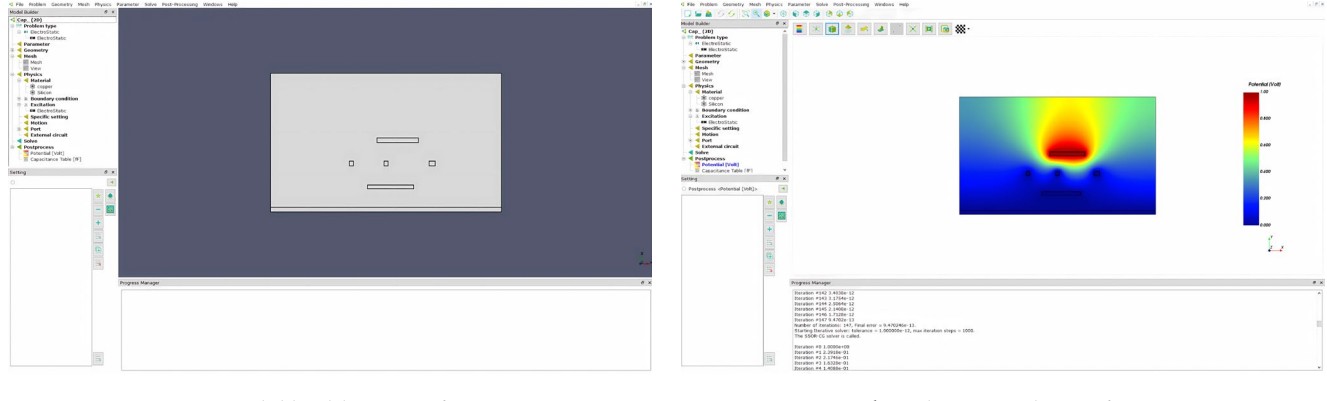

(**a**) Model building interface          (**b**) Solving result interface

**Figure 4.** The model building and solving result interfaces of EMPbridge.

### 2.3. Data Augmentation

The size of datasets also has a crucial impact on the deep learning prediction effect. When the size is small, due to a lack of sufficient supervision information, it is easy to lead to poor generalization ability of the model, and the phenomenon of over-fitting is prone to occur [26]. However, the acquisition of enough datasets, such as with the FEM simulation, is often accompanied by heavy time and labor costs. Data augmentation is a way to alleviate this problem. It can expand the original data by setting a transformation rule without additional computation. Common transformation operations include rotation, mirror flip, cropping, etc. [27].

This work introduces a mirror flip method, as shown in Figure 5, to augment datasets. After flipping, the geometric variables of conductors change as follows. The position and width of conductor 1 are unchanged; The widths of conductors 4 and 5 are unchanged, but their horizontal positions are multiplied by $-1$; For conductors 2 and 3, their widths are directly exchanged, and horizontal coordinates are first multiplied by $-1$ and then exchanged. The transformation rule of parasitic capacitances is shown in Figure 6. $C$ represents the original value before the mirror flip, and $\hat{C}$ denotes the value of the new capacitance matrix.

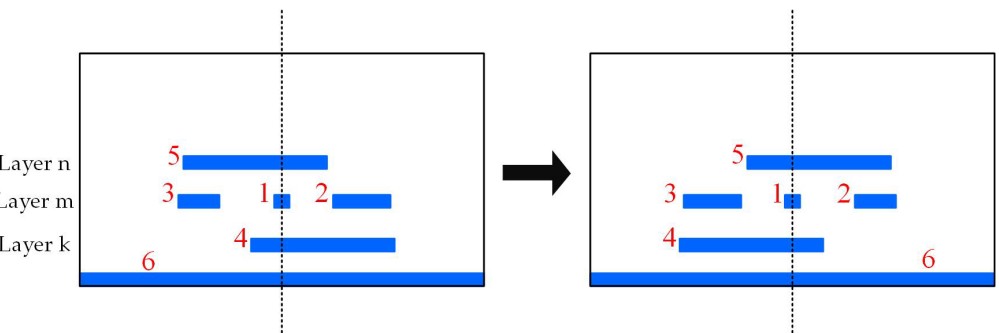

**Figure 5.** The mirror flip illustration of the studied pattern.

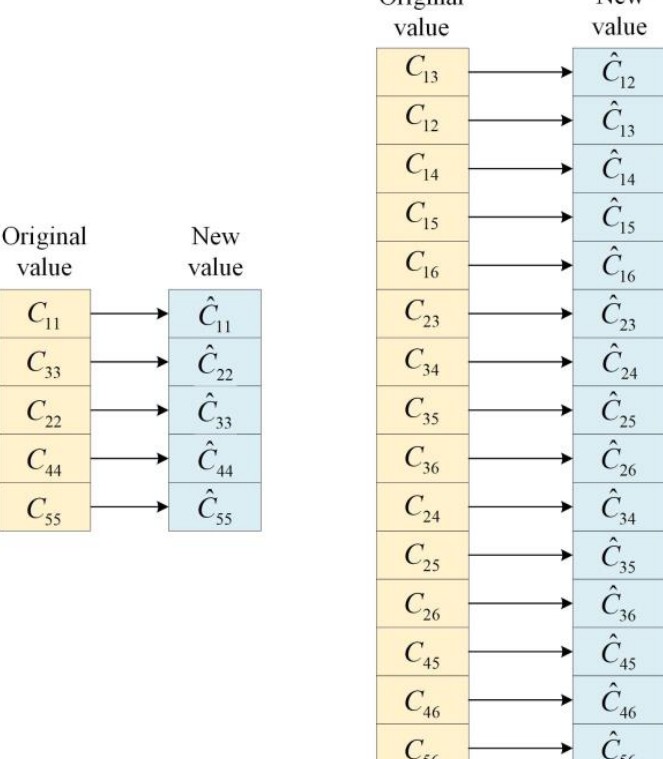

**Figure 6.** The transformation rule between the original data and the new data.

According to the transformation rule, a new set of data can be obtained, and the datasets are doubled in size. It is worth remembering that this data augmentation method is also applicable to other similar patterns once a transformation rule is determined.

### 2.4. Prediction of Parasitic Capacitance Matrix Based on the DNN

In the parasitic capacitance matrix, the total capacitance equals the sum of the remaining coupling capacitances in the same row. With this information, the DNN network output includes only coupling capacitances in this work, and the total capacitance is indirectly obtained by summing the predicted coupling capacitances. The input of DNN is a vector containing geometric variables, and the output is a vector containing all coupling capacitances.

Taking the studied pattern as an example, its parasitic capacitance matrix has been given in Equation (3). Due to the matrix's symmetric property, there are 15 non-repetitive coupling capacitances in total, which are $C_{12}$, $C_{13}$, $C_{14}$, $C_{15}$, $C_{16}$, $C_{23}$, $C_{24}$, $C_{25}$, $C_{26}$, $C_{34}$, $C_{35}$, $C_{36}$, $C_{45}$, $C_{46}$, and $C_{56}$. Therefore, as shown in Figure 7, the input dimension of DNN is nine, that is, all geometric variables; the output dimension is fifteen, including all coupling capacitances. The batch normalization (BN) layer could mitigate the vanishing gradient problem since it shifts the input data distribution to the unsaturated region of the activation function [28]. Hence, we add a BN layer before each activation function in the DNN structure, although it is not shown in Figure 7. Equation (4) is the loss function, which is defined by the mean square error (*MSE*) between the predicted value and the reference value in the datasets. In Equation (4), *N* represents the size of datasets, and *K* denotes the output dimension of the DNN, which is fifteen here. $C_{ij}$ is the predicted value, and $C'_{ij}$ represents the reference value.

$$MSE = \frac{1}{N}\sum_{i=1}^{N}\frac{1}{K}\sum_{j=1}^{K}\left|C_{ij} - C'_{ij}\right|^2 \tag{4}$$

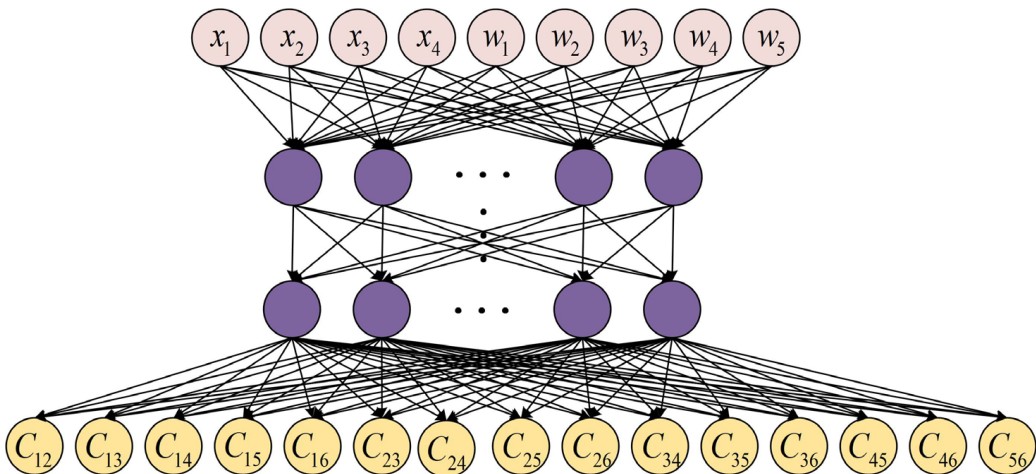

**Figure 7.** The illustration of DNN structure.

### 2.5. Density-Based Data Representation

For comparison, we employ the convolutional neural network ResNet to predict the capacitance matrix. A density-based data representation method [11,29] is used to describe the geometric characteristics of the metal layer and represents the input of ResNet. This method can effectively describe the layout feature [23]. Furthermore, it is suitable for the feature extraction operation performed by the convolutional neural network. In this way, the conductor placement of a metal layer can be represented as a vector.

The extraction window width *W* determines the extraction range of the pattern in the horizontal direction and the variation range of the geometric variables. A simulation experiment is conducted to determine the extraction window width *W* of the studied pattern. In the simulation, conductor 1 and conductor 3 take the minimum width under the process. Conductor 4 and conductor 5 take a very large width. Then move conductor 3 away from conductor 1 until the coupling capacitance between them is less than 1% of the total capacitance of conductor 1. The distance between conductor 1 and conductor 3 at this time multiplied by two is the width of the extraction window [5,11].

Once the extraction window size *W* is determined, the flow of this data representation method is as follows. First, the extraction window size is divided into *L* cells at equal intervals. To accurately distinguish different geometries, the cell width should not be more than half the minimum spacing $s_{min}$ between two interconnects [11]. Therefore, in this study, we let the cell width equal half the minimum spacing, and the number of cells *L* is determined by Equation (5) [23].

$$L = \frac{W}{0.5 \times s_{min}} \tag{5}$$

Second, taking one layer as an example, a one-dimensional vector with a size of *L* is created. The vector value is determined by the ratio of the conductor area to the cell area at each position. When the conductor occupies the entire cell area, the vector value is 1; when the conductor occupies part of a cell, the vector value is taken as the ratio of the occupied area to the cell area. The illustration of the density-based data representation of one layer is shown in Figure 8.

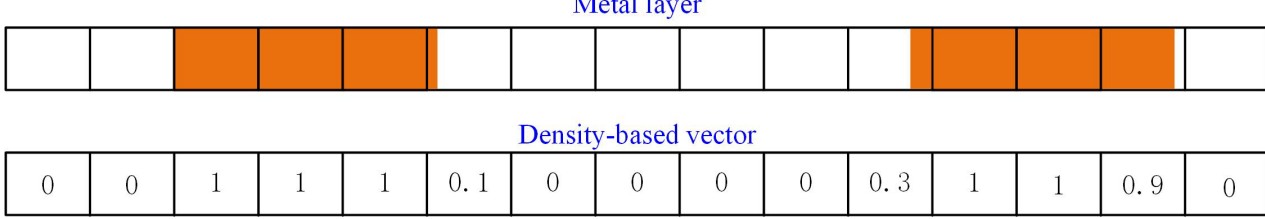

**Figure 8.** The density-based data representation of one metal layer.

### 2.6. Prediction of Parasitic Capacitance Matrix Based on ResNet

ResNet was proposed by He [30] in 2015, which solves the network degradation problem of deep networks. In this study, we modify the convolution kernel size of ResNet to fit our input data and then use it to predict the parasitic capacitance matrix. The network structure ResNet-18 used in this work is shown in Figure 9, including 17 two-dimensional convolutional layers, one fully connected layer, and two pooling layers. "$1 \times 7$ conv, 64, /2" in the figure indicates that the output channel of the convolutional layer is 64, the size of the convolution kernel is $1 \times 7$, and the stride is 2. The connection line in the figure is called a shortcut. The solid line indicates that the input and output of the residual module are directly connected. The dotted line means the input and output of the residual module are inconsistent in size and cannot be directly connected. Hence a $1 \times 1$ convolutional kernel is added, which makes the input and output the same size. A detailed illustration of these two shortcuts is also given in Figure 9. The loss function definition of the ResNet is the same as Equation (4).

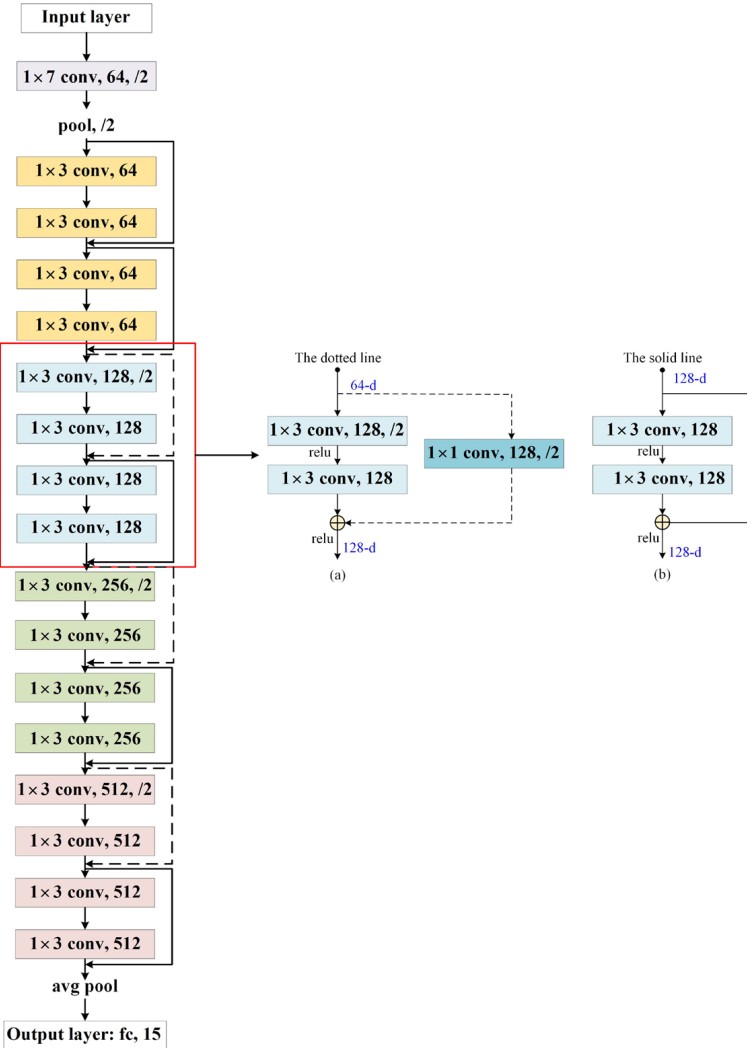

**Figure 9.** The neural network structure of the modified ResNet-18. (a) The concrete structure of the dotted line; (b) The concrete structure of the solid line.

## 3. Experiments and Results

The research case in this experiment has been introduced in Section 2.2. It has nine geometric variables. Equation (3) defines its parasitic capacitance matrix, which includes fifteen non-repetitive coupling capacitances. In this study, only the coupling capacitances are considered during the deep learning training, and the total capacitance is calculated by summing up coupling capacitances. The structure and geometric variables' sampling range of the case meet the 55 nm process requirements. Moreover, this work is also applicable to other process technology. Table 2 lists the first four layers' process criterion of the 55 nm, which includes the interconnect wires' thickness, the minimum wire width $w_{min}$, and the minimum spacing $s_{min}$ between two wires of each metal layer. This research selects the combination of layer 2, layer 3, and layer 4, that is, $k = 2$, $m = 3$, and $n = 4$ in Figure 2.

**Table 2.** Multilayer interconnect 55 nm process standard (first four metal layers) [11].

| Layer | Thickness (μm) | $w_{min}$ (μm) | $s_{min}$ (μm) |
|---|---|---|---|
| 1 | 0.1 | 0.054 | 0.108 |
| 2 | 0.16 | 0.081 | 0.08 |
| 3 | 0.2 | 0.09 | 0.09 |
| 4 | 0.2 | 0.09 | 0.09 |

As shown in Figure 10, the extraction window width is taken as $W = 56 \times s_{min}$, where $s_{min} = 0.09$ μm. For a higher precision in reference values, the solving domain width of numerical computation is taken as 10 μm. Once the window width $W$ is determined, the ResNet input data dimension $L$ of one metal layer can be computed using Equation (5), and the result is $L = 112$. Since there are three metal layers, the input data size of the ResNet is $3 \times 112$.

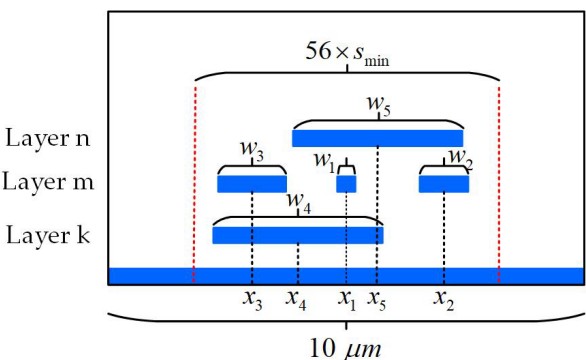

**Figure 10.** The illustration of extraction window width and the solving domain width.

Traditional multilayer interconnect routing follows the Manhattan structure [31], that is, placing adjacent layers' interconnects orthogonally to each other. In this way, the crosstalk caused by coupling capacitances can be minimized [32]. The studied pattern shown in Figure 2 is a cross-section view of a small geometry from the layout. Layers $k$ and $n$ are regarded as horizontal routing here, and the three conductors in the middle layer are regarded as vertical page routing. $w_1$, $w_2$, and $w_3$ represent the width of the interconnects. Hence, we let the maximum variation range of $w_1$, $w_2$, and $w_3$ not exceed ten times the minimum width. Finally, the sampling range of each geometric variable is listed in Table 3.

**Table 3.** Parameters' sampling range.

| Parameter | Sampling Range | Parameter | Sampling Range |
|:---:|:---:|:---:|:---:|
| $x_2$ | [1.0, 2.05] | $w_2$ | [0.09, 0.9] |
| $x_3$ | [−2.05, −1.0] | $w_3$ | [0.09, 0.9] |
| $x_4$ | [−1.0, 1.0] | $w_4$ | [0.081, 3.0] |
| $x_5$ | [−1.0, 1.0] | $w_5$ | [0.09, 3.0] |
| $w_1$ | [0.09, 0.9] | - | - |

The deep learning framework used in the study is PyTorch. The hardware configuration is as follows: CPU Intel Core i7-12700KF, GPU NVIDIA RTX3080Ti, with 16GB RAM. For the studied pattern, 25,000 sets of samplings were generated, and the reference capacitance matrix values were computed using the FEM solver using EMPbridge software. Finally, 50,000 sets of data were obtained after using the data augmentation method described in Section 2.2. The total datasets were randomly split into three parts: 80% of the sampling was used as the training set, 10% was used as the validation set, and 10% was used as the testing set. Too-small coupling capacitances are usually not considered in the extraction [11,23]. In this work, all coupling capacitances are considered during the neural network training stage. However, in the testing stage, the coupling capacitances whose values are less than 1% of the corresponding total capacitance are specially treated. When calculating the predicted total capacitance, all coupling capacitances are included in the calculation. When evaluating the prediction accuracy of the coupling capacitances, the small values are not considered in order to evaluate only the accuracy of significant coupling capacitances. Since one coupling capacitance corresponds to two total capacitances, such as $C_{ij}$, which contributes to total capacitances $C_{ii}$ and $C_{jj}$, the error of $C_{ij}$ is not evaluated only when both $C_{ij}/C_{ii}$ and $C_{ij}/C_{jj}$ are less than 1%.

### 3.1. Experiment Results

The hyperparameters of the neural network have a great influence on the prediction effect. Different combinations of hyperparameters may cause a significant difference in the prediction effect of the model. For scenarios with few kinds of hyperparameters, the grid search method is commonly used. However, the DNN has many hyperparameters needed to be tuned, and the grid search method will become time-consuming. To alleviate this problem, we utilize an automated hyperparameter tuning method, the tree-structured parzen estimators (TPE) method [33]. TPE is based on the Bayesian optimization algorithm, and it can select the next set of hyperparameters according to the evaluation results of the existing hyperparameter combinations [34]. The TPE method can find the better or best choice among all hyperparameter combinations within the maximum evaluation number.

For the DNN training, the optimizer is Adam [35]. Weights and biases are initialized by the Xavier initialization [36]. The remaining hyperparameters are determined with TPE, and their search space is shown in Table 4. The maximum evaluation number is 100. The objective function of the TPE is the predicted coupling capacitances' average relative error of the trained neural network on the testing set, and the procedure is implemented using the Hyperopt python package [37]. In the end, the hyperparameter combination with better performance for this studied pattern is as follows. The activation function is Swish; the batch size is 32; the learning rate is $10^{-4}$; the number of hidden layers is 4, and each hidden layer has 500 neurons.

**Table 4.** Hyperparameters' space for TPE.

| Hyperparameter | Search Range |
|---|---|
| Activation function | ReLU [38], Tanh, Swish [39] |
| Batch size | 32, 64, 128, 256, 512 |
| Learning rate | $10^{-2}, 10^{-3},\ 10^{-4}, 10^{-5}$ |
| Number of hidden layers | 1, 2, 3, 4, 5, 6 |
| Neurons in each hidden layer | 50, 100, 200, 300, 400, 500, 600 |

Additionally, we experimented with ResNet-18 and ResNet-34 to predict their respective capacitances. Both employ the Adam optimizer. Since their hyperparameters needed to be tuned only for batch size and learning rate, hence we utilized the grid search method to find the better combination. Batch size is enumerated in {32,64,128,256,512}; the learning rate is enumerated in {$10^{-2}, 10^{-3}, 10^{-4}, 10^{-5}$}. Eventually, ResNet-18 has better performance than ResNet-34, and its hyperparameter combination is as follows. The batch size is set to 32, and the learning rate is $10^{-3}$.

Tables 5 and 6 are the prediction results obtained by those two neural networks under 1000 epochs, and their testing sets are consistent. The predicted total capacitances are indirectly obtained by summing the predicted coupling capacitances. We can see that DNN performs better both in the coupling capacitance and total capacitance prediction in this research case. Furthermore, the training time of DNN is only 31% of ResNet-18. In addition, those two networks' prediction accuracy on the total capacitance all meets the accuracy requirements (<5%) in the advanced process criterion, showing the computation method of total capacitance introduced in this work is reliable.

**Table 5.** The predictive performance of different neural networks on the coupling capacitance (the relative error is only evaluated for coupling capacitances greater than 1% of the corresponding total capacitance).

| Network | Training Time (h) | Average Relative Error | Maximum Relative Error | Ratio (>5%) |
|---|---|---|---|---|
| DNN | 0.89 | 0.12% | 10.66% | 0.01% |
| ResNet-18 | 2.90 | 0.21% | 10.69% | 0.05% |

**Table 6.** The predictive performance of different neural networks on the total capacitance.

| Network | Average Relative Error | Maximum Relative Error | Ratio (>5%) |
|---|---|---|---|
| DNN | 0.05% | 2.76% | 0 |
| ResNet-18 | 0.05% | 4.45% | 0 |

Figure 11 shows the relative error distribution of two neural networks for the coupling capacitance and the total capacitance, respectively. The larger errors of coupling capacitance usually occur at lower capacitance values. In addition, the total capacitance error distribution of DNN is more concentrated than that of ResNet-18.

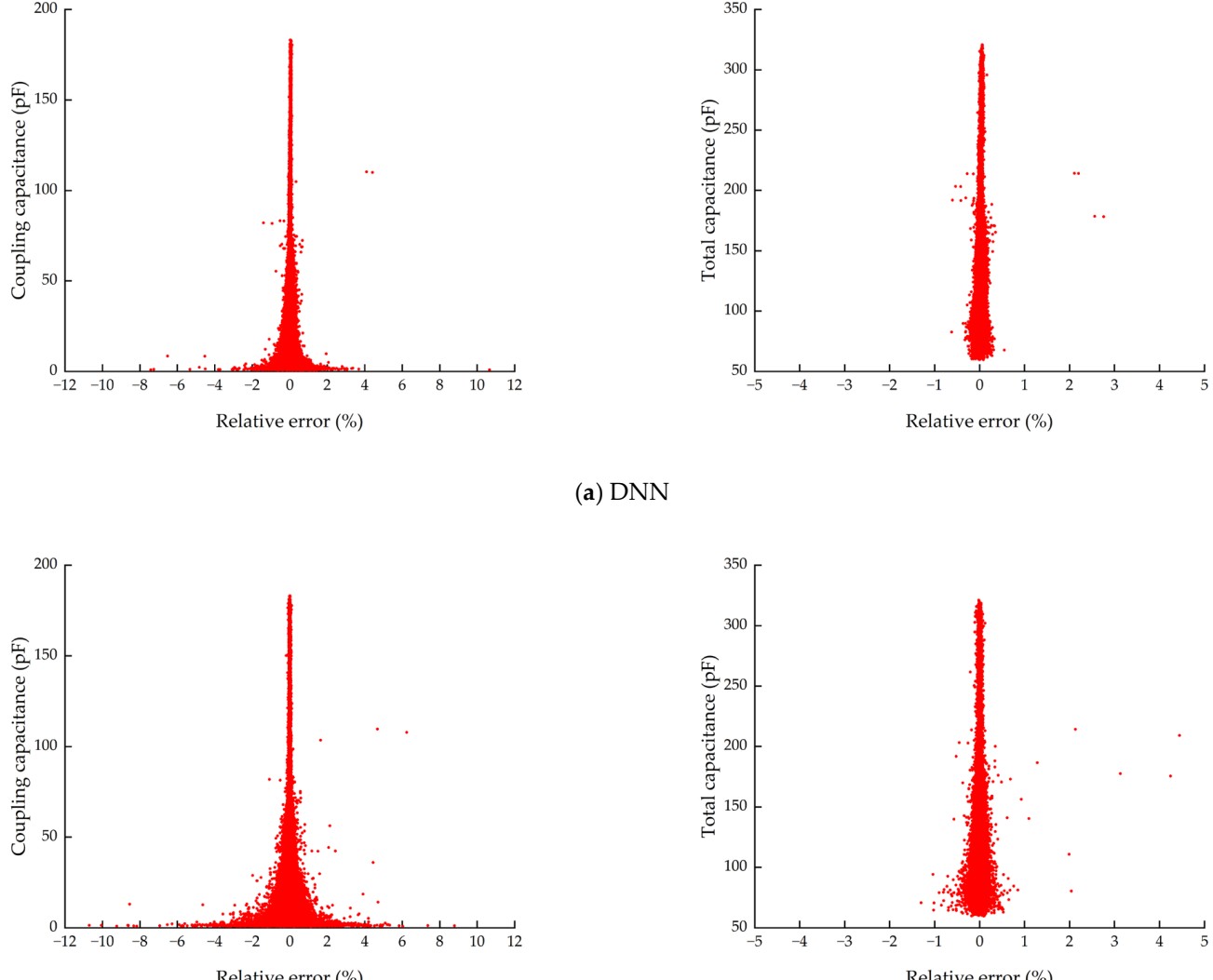

**Figure 11.** The relative error distribution of the predicted coupling capacitance and total capacitance: (**a**) DNN; (**b**) ResNet-18.

### 3.2. The Comparison of Two Methods for Predicting Total Capacitance Using the DNN

In this study, the total capacitance is obtained indirectly by calculating the sum of coupling capacitances. To compare this method with the way that directly predicts the total capacitance using DNN, we constructed another DNN model called DNN-2. The input of this neural network is consistent with the DNN structure introduced in Section 2.4, but

the output dimension of DNN-2 is five, corresponding to the five total capacitances of this research case. The training sets, validation sets, and testing sets are consistent with those in Section 3.1. The approach of hyperparameters' settings of DNN-2 is the same as those of the DNN introduced in Section 3.1.

Finally, after trial and error, the hyperparameters' combination with better performance is as follows. The activation function is Tanh. The learning rate is $10^{-3}$. The batch size is 64. The number of hidden layers is 3, and the number of neurons in each hidden layer is 500. The prediction results of DNN-2 are shown in Table 7. At the same time, the prediction results of DNN on the total capacitance in Section 3.1 are given as a comparison. From the table, we can see that the predictive performance of the two DNN models is very close. Therefore, the method used in this study is reliable. Moreover, the result also indicates that our method can save the training cost for predicting total capacitance.

**Table 7.** The prediction results of two methods using DNN for total capacitance.

| Method | Training Time (h) | Average Relative Error | Maximum Relative Error | Ratio (>5%) |
|--------|-------------------|------------------------|------------------------|-------------|
| DNN | 0.89 | 0.05% | 2.76% | 0 |
| DNN-2 | 0.35 | 0.05% | 4.96% | 0 |

*3.3. The Predictive Performance of DNN under Different Training Set Sizes*

This subsection tests the predictive performance of DNN under different training set sizes. The validation set and testing set are still the same as in Section 3.1. The ratio of the training set size gradually increases from 10% to 80%; that is, the size of training sets varies from 5000 to 40,000. The structure of DNN is also the same as in Section 3.1. Finally, the average relative error and the ratio of errors greater than 5% of each trained model on the testing set are drawn in Figure 12. It can be seen that both the average relative error and the ratio (>5%) overall decrease with the increase of the training set size. Moreover, when the size of training sets is more than 20,000, the change trend in both factors slows down.

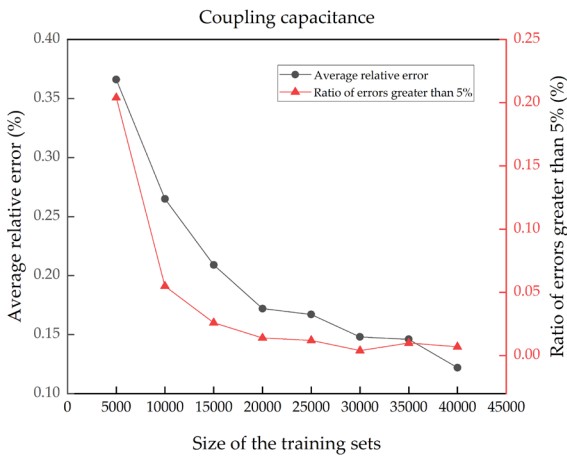

(**a**) Coupling capacitance

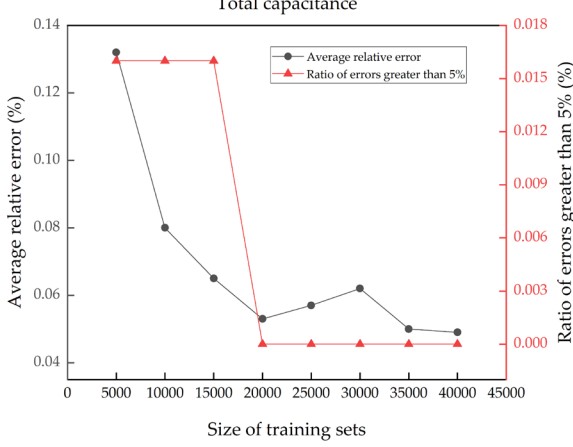

(**b**) Total capacitance

**Figure 12.** Distribution of average relative error and the ratio of errors greater than 5% under different sizes of training sets: (**a**) Coupling capacitance; (**b**) Total capacitance.

*3.4. The Comparison of Solving Time between the FEM and the Trained DNN*

To compare the solving efficiency between the trained DNN and the FEM, we select 100 sets of data in the testing set and compute their parasitic capacitance matrix with the trained DNN and FEM, respectively. Since both the trained DNN and FEM need to process the input data in advance (i.e., FEM needs to generate mesh, and the trained DNN needs

to construct the input vector), the data preparation time is ignored here, and only the solving time is counted. The programming language of DNN is Python, and it uses GPU for computation. The programming language of the FEM is C++. Finally, the average computation time of those two methods for one case is shown in Table 8. The solving time of the trained DNN takes only 2% of FEM.

**Table 8.** The average computation time of the FEM and the trained DNN for one testing case.

| Method | Time (ms) |
|---|---|
| FEM | 759.49 |
| The trained DNN | 14.64 |

*3.5. The Verification of a New Pattern*

To further verify our method, we experiment with a new pattern called pattern-2, whose geometry and parameters are shown in Figure 13. In pattern-2, conductor 1 is centered, conductor 2 and conductor 4 are always on the right side of the center line, and conductor 3 and conductor 5 are always to the left of the center line. We choose the same metal layers as the pattern in Figure 2, but each layer contains a different number of conductors. Subsequently, we built a new DNN network called DNN-3. Since the geometric variables and the total number of conductors of pattern-2 are the same as the pattern in Figure 2, the input and output layer used for DNN-3 is the same as in Figure 7.

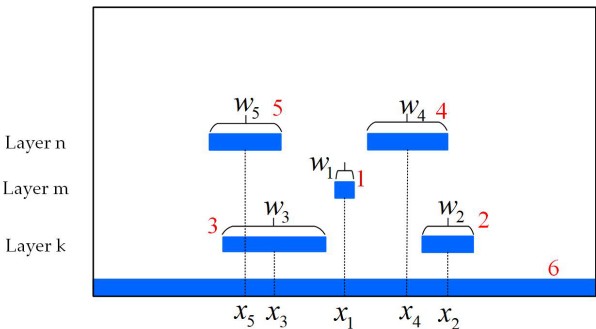

**Figure 13.** The illustration of pattern-2 and its geometry variables.

The metal layer selected in Pattern-2 is the same as the pattern in Figure 2; hence their extraction window width is also the same. The sampling range of each geometric parameter in pattern-2 is shown in Table 9. We utilized the field solver EMPBridge to generate 25,000 sets of data and used the data augmentation method introduced in Section 2.3 to augment the datasets, and finally obtained 50,000 sets of data. A total of 80% of the datasets were used as training sets, 10% were used as verification sets, and 10% were used as testing sets.

**Table 9.** Parameters' sampling range of pattern-2.

| Parameter | Sampling Range | Parameter | Sampling Range |
|---|---|---|---|
| $x_2$ | [1.04, 1.52] | $w_2$ | [0.081, 2.0] |
| $x_3$ | [−1.52, −1.04] | $w_3$ | [0.081, 2.0] |
| $x_4$ | [1.05, 1.52] | $w_4$ | [0.09, 2.0] |
| $x_5$ | [−1.52, −1.05] | $w_5$ | [0.09, 2.0] |
| $w_1$ | [0.09, 0.9] | - | - |

The approach of hyperparameters' settings of DNN-3 is the same as those of the DNN introduced in Section 3.1. In the end, the better hyperparameter combination obtained with TPE is as follows. The activation function is Swish. The batch size is 64. The learning rate

is $10^{-4}$; The number of hidden layers is 2, and each hidden layer has 500 neurons. The prediction performance of DNN-3 on the testing set is shown in Table 10. It can be seen that our method performs well in the new example, and the average relative error of the coupling capacitance is only 0.10%, the average relative error of the total capacitance is only 0.04%, and the prediction accuracy of the total capacitance all meets advanced technology accuracy requirements (errors < 5%).

**Table 10.** The predictive performance of DNN-3 on the coupling capacitances and total capacitances.

| Property | Training Time (h) | Average Relative Error | Maximum Relative Error | Ratio (>5%) |
|---|---|---|---|---|
| Coupling Capacitance | 0.32 | 0.10% | 5.04% | 0.00% |
| Total capacitance | - | 0.04% | 0.37% | 0 |

## 4. Discussion

This study uses the DNN to predict the parasitic capacitance matrix of a 2D pattern. The network output only includes coupling capacitances. The total capacitance is obtained by calculating the sum of the corresponding coupling capacitances. Moreover, a data augmentation method is employed, doubling the dataset's size. For comparison, this study also utilized ResNet to predict the parasitic capacitance matrix. The above experiments can derive the following discussions.

- Data augmentation technique can obtain new data through the original dataset without additional computation. Section 2.2 introduces a mirror flip method that doubles the size of datasets size and reduces the data preparation effort.
- In this paper, we train only one DNN model to predict the capacitance matrix. The neural network's output includes only coupling capacitances in the capacitance matrix, and the total capacitances are obtained by summing corresponding predicted coupling capacitances. For the research case, experimental results in Table 6 show that the prediction accuracy of the total capacitance obtained by this method meets the advanced process requirements (<5%).
- The comparison in Table 7 indicates that the predictive performance of our method with the method that directly predicts the total capacitance using DNN is very close. Moreover, the training time of our method is 0.89 h, and that of DNN-2 is 0.35 h. Therefore, our method only takes about 72% of the total training time but can simultaneously predict the total capacitances and coupling capacitances. In addition, the hyperparameters' tuning of DNN-2 is time-consuming, and our method also saves this process. The above results indicate that the method introduced in this work is reliable and can save the training cost for predicting total capacitance.
- In [11], ResNet is used to predict the total capacitance and coupling capacitances, which shows good precision. To compare this kind of neural network's prediction performance with DNN for this research case, we also utilized the ResNet to predict the capacitance matrix. The results show that the performance of DNN is better than the ResNet for this research case, and the training time of DNN is only 31% of ResNet-18.
- We tested the prediction effect of DNN under different dataset sizes. With the increase in training data size, the average relative error and the ratio of errors greater than 5% of trained models gradually decrease. Further, when the training data size is greater than 20,000, the changing trend of the average relative error and the ratio (>5%) tends to slow down.
- In addition, the calculation efficiency of the FEM and the trained DNN are compared. The average computing time of the trained DNN for one case is only 2% of that of FEM, which shows that the trained DNN has good efficiency. Furthermore, we validate our method to another pattern called pattern-2. The results show that our method performs well on the new pattern, indicating its feasibility.

- This experiment only considers horizontal changes in the interconnect geometry. However, in real manufacturing, there will be more process variations. For example, the permittivity of the dielectric layer, interconnect wire's thickness, etc. Therefore, in the future, the parasitic capacitance prediction will add more variables to the consideration.
- When there are corners or connections in the interconnect wires [4,40], the two-dimensional analysis is no longer applicable. We prepare to solve the parasitic capacitance prediction of this kind of problem in the next step. The neural network's input data representation is critical, and one approach is using the density-based data representation of the interconnect's top view [23].

### 5. Conclusions

This study used DNN to predict the parasitic capacitance matrix of a two-dimensional pattern. The neural network's output includes only coupling capacitances in the capacitance matrix, and the total capacitances are obtained by summing corresponding coupling capacitances, thus saving the cost of training total capacitance. For the research case in this study, experimental results show that the average relative error of the predicted total capacitance based on DNN in the testing set is only 0.05%, and the average relative error of the predicted coupling capacitance is only 0.12%, showing good prediction accuracy. Furthermore, a data augmentation method was introduced, reducing the data preparation effort. Our study can be used to construct capacitance models in the pattern-matching method.

**Author Contributions:** Conceptualization, Y.M., X.X., S.Y. and Z.R.; Data curation, Y.M., Y.Z. and T.Z.; Formal analysis, Y.M., X.X., S.Y. and Z.R.; Funding acquisition, X.X., S.Y. and Z.R.; Investigation, S.Y. and L.C.; Methodology, Y.M., X.X. and S.Y.; Project administration, Z.R.; Resources, Y.M., X.X., Y.Z., T.Z. and Z.R.; Software, Y.M., S.Y., Y.Z., T.Z. and Z.R.; Supervision, X.X., S.Y., Z.R. and L.C.; Validation, Y.M.; Visualization, Y.M.; Writing—original draft, Y.M.; Writing—review & editing, X.X., S.Y., and Z.R. All authors have read and agreed to the published version of the manuscript.

**Funding:** This research was funded by The Institute of Electrical Engineering, CAS (Y960211CS3, E155620101, E139620101).

**Data Availability Statement:** The data that support the findings of this study are available from the corresponding author upon reasonable request.

**Conflicts of Interest:** The authors declare no conflict of interest.

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
