# Peer review of "Extraction of Interconnect Parasitic Capacitance Matrix Based on Deep Neural Network"

_electronics, doi:10.3390/electronics12061440_

Round 1
Reviewer 1 Report
This paper presents a methodology that uses DNN to predict the interconnect parasitic capacitance matrix of a two-dimensional pattern.
The manuscript is quite well written and the basic idea is interesting.
However, it is not clear how the groundthrouth is achieved.
Furthermore, further examples of circuit layout and comparisons with the state of the art are to be provided.
Reviewer 2 Report
1. The objective of the work is novel, the authors have explained in a neat manner
2. But the list of contributions from lines 101 to 110 cannot be considered as contributions as they are part of the implementation step and the contributions will not stand alone as significant contributions.
3. Authors can develop a few models on their own and compare with pre-trained models.
4. If there are any work in literature, the proposed method can be compared with the existing work and their proposed method results, importance can be highlighted
Reviewer 3 Report
An interesting topic is taken under study by the authors. Application of data driven surrogate models for modeling of expansive simulation problems is one of the topics that actively being persuaded by many researchers. The selected topic is a good example and a worthy topic for the journals readers. However the work requires some revisions for acceptance.
The counterpart methods must be extended with more examples such as Simple traditional ANN (MLP), SVRM, Ensemble and other algorithms.
It is a well-known fact that hyper-parameter adjustment is a crucial step where a small error might end up designer with a very poor modeling performance. How authors did determined the hyper-parameters (layer size and hidden neuron numbers, activation functions etc.) of their model? Authors should check literature for automated deep learning works where the hyper-parameters are obtained via optimization techniques. And also how authors did ensured that their model did not fall into over-fitting, it would be good to use a portion of the data as should out and then determine the optimal hyper-parameter using an optimization process [A1]. Can be considered as an example for this mean where modeling of microwave transistors is taken as a data driven surrogate modeling problem.
[A1] Calik, N., GüneÅŸ, F., Koziel, S. et al. Deep-learning-based precise characterization of microwave transistors using fully-automated regression surrogates. Sci Rep 13, 1445 (2023). https://doi.org/10.1038/s41598-023-28639-4
What would be the performance of the model in different training-validation-test/hold-out sets? For an example (50%-20%30%) ratio or by reducing the total number of samples. It is obvious that this would reduce the accuracy, but it would also a good example to present the generalization capability of the proposed approach.
The literature overview of the work is weak. There is only 2 cited work from 2022 and 3 form 2021. Authors must extend their literature overview with more recently published works. Authors are encouraged to use following or similar works form literature to extend their literature overview.
https://doi.org/10.1007/s00521-021-06667-3
https://doi.org/10.1016/j.energy.2021.122955
https://doi.org/10.1109/ACCESS.2023.3243132
https://doi.org/10.1016/j.chemolab.2022.104520
https://doi.org/10.1016/j.autcon.2022.104672
https://doi.org/10.1016/j.egyai.2022.100214
Round 2
Reviewer 1 Report
The authors have addressed enough of my comments and suggestions.
In addition, the authors further improved this manuscript by answering other reviewers’ questions and revising the manuscript.
Consequently, this paper is now acceptable for publication.
Reviewer 3 Report
Authors had made an effort to improve the overall quality of the work. The reviewer has no furthered comments for this work. The current form of the work can be accepted for publication.